# Safety and Efficacy of Nusinersen and Risdiplam for Spinal Muscular Atrophy: A Systematic Review and Meta-Analysis of Randomized Controlled Trials

**DOI:** 10.3390/brainsci13101419

**Published:** 2023-10-07

**Authors:** Yue Qiao, Yuewei Chi, Jian Gu, Ying Ma

**Affiliations:** Department of Neurology, Shengjing Hospital of China Medical University, Shenyang 110055, China

**Keywords:** spinal muscular atrophy, nusinersen, risdiplam, meta-analysis, efficacy, safety

## Abstract

Objective: We performed a systematic review and meta-analysis of the efficacy and safety of nusinersen and risdiplam in the treatment of spinal muscular disease (SMA). Methods: We screened the literature published in Pubmed, Web of Science, Embase, and Cochrane before July 2023 to conduct randomized controlled trials to test the treatment of SMA patients with nusinersen and risdiplam. The data were analyzed using Review Manager 5.4 software and Stata version 15.0 software. Results: A total of six randomized controlled trials were included, involving 728 SMA patients, to synthesize evidence. It is reported that nusinersen treatment was beneficial for increasing the score of the Hammersmith Functional Motor Scale—Expanded (HFMSE) (WMD: 4.90; 95% CI: 3.17, 6.63; *p* < 0.00001), Revised Upper Limb Module (RULM) (WMD: 3.70; 95% CI: 3.30, 4.10; *p* < 0.00001), and Hammersmith Infant Neurological Evaluation Section 2 (HINE-2) (WMD: 5.21; 95% CI: 4.83, 5.60; *p* < 0.00001). In addition, the risdiplam treatment group also showed statistically significant improvements in the HFMSE score (WMD:0.87; 95% CI: 0.05, 1.68; *p* = 0.04), the 32-item Motor Function Measure (MFM32) (WMD:1.48; 95% CI: 0.58, 2.38; *p* = 0.001), and (WMD: 1.29; 95% CI: 0.57, 2.01; *p* = 0.0005). Nusinersen and risdiplam did not cause a statistically significant increase in the RULM score for adverse events (OR: 0.93; 95% CI: 0.51, 1.7; *p* = 0.82) and for severe adverse events (OR: 0.77; 95% CI: 0.47, 1.27; *p* = 0.31). Conclusion: Our analysis found that nusinersen and risdiplam treatment showed clinically meaningful improvement in motor function and a similar incidence rate of adverse events compared with the placebo. Further research should be carried out to provide a direct comparison between the two drugs in terms of safety and efficacy.

## 1. Introduction

Spinal muscular atrophy (SMA) is a debilitating hereditary neuromuscular disorder characterized by progressive muscle weakness and atrophy, hypotonia, and respiratory distress [1]. The clinical features of SMA vary in degrees of severity but all patients have in common progressive muscle weakness and mobility impairment [2]. It is a recessive genetic disease that represents the second leading cause of infancy and early childhood death, with an incidence of around 1:10,000 live births and an average carrier frequency that varies from 1:40 to 1:60 worldwide [3].

SMA is caused by loss-of-function mutations of the *SMN1* (survival of motor neuron 1) gene located on chromosome 5q13, and more than 95 percent of SMA patients show homozygous deletion of exon 7 of the *SMN1* gene, leading to a reduction in survival motor neuron (SMN) protein and motor neuron dysfunction [4]. The other version of SMN is a highly homologous gene, *SMN2*, which partially functionally compensates for an *SMN1* homozygous deletion. The only functional difference between the two SMN genes is a synonymous transition leading to alternative splicing of *SMN2* exon 7 and non-functional protein (SMN-Δ7) instead of the full-length protein. Previous clinical studies showed that the *SMN2* copy number is negatively correlated with disease severity and a higher level of functional SMN protein is associated with milder phenotypes [5]. According to onset age and motor milestones, SMA has generally been classified into five phenotypes (type 0 to type IV) [6]. SMA type I is the most common phenotype, occurring in the first six months of life, and it is associated with functional impairments and progressive weakness of the respiratory muscles, leading to a high rate of infant mortality or invasive ventilation by 2 years of age [7,8].

Due to SMA resulting from low levels of SMN protein, new treatment strategies mainly revolve around methods of increasing SMN protein levels through modifying *SMN2* gene splicing or gene replacement therapy [9]. In recent years, several novel disease-modifying drugs, including nusinersen, risdiplam, and zolgensma, have been developed and significantly changed the course of the disease. In December 2016, nusinersen, an antisense oligonucleotide drug, was the first potential therapy approved by the US Food and Drug Administration (FDA) to treat SMA [10]. Nusinersen (Spinraza) alters the splicing of *SMN2* pre-mRNA and increases the production of SMN protein [11], and the treatment requires four loading doses on days 0, 14, 28, and 64 and maintenance doses every four months through intrathecal injection [12,13]. Later, another gene replacement therapy named zolgensma (onasemnogene, AVXS-101), which delivers the *SMN1* gene to motor neurons and produces full-length functional SMN protein through the adeno-associated virus (AAV), was approved by the FDA in May 2019 [14,15]. Additionally, risdiplam (Evrysdi) was recently approved for the treatment of patients 2 months old and older. It is a small-molecule *SMN2*-splicing modifier that improves the efficiency of the transcription of the *SMN2* gene, thus increasing the systemic SMN protein concentration [16,17]. Because of the various clinical features and rarity of SMA, there have been limited randomized controlled trials (RCTs) conducted to evaluate the efficacy and safety of these drugs.

Recently, there has been growing interest in these disease-modifying drugs because of the great progress in the treatment of this severe neurodegenerative disease. Investigators of nusinersen trials reported that it is possible to improve disease symptoms rather than only slow down disease progression [18,19]. At the same time, risdiplam was found to dramatically improve patient survival through increasing functional SMN protein and provided convenience to patients as it can be orally administered [20,21]. Many clinical trials have shown that nusinersen and risdiplam improve disease symptoms, but there is presently no cure for SMA.

The purpose of this study was to conduct a systematic review and meta-analysis to evaluate the safety and effectiveness of nusinersen and risdiplam in the treatment of SMA patients.

## 2. Methods

### 2.1. Literature Search

We performed this systematic review and meta-analysis following the PRISMA (Preferred Reporting Item for Systematic Review and Meta-Analysis) 2020 statement [22] and searched PubMed, Embase, Web of Science, and Cochrane through 9 July 2023. This study was registered in the Prospective Register of Systematic Reviews (CRD42023454263). We used combinations of the following keywords for the literature search: “spinal muscular atrophy”, “SMA”, “nusinersen”, and “risdiplam”. The search strategy is described in the Appendix A. Two independent researchers screened the titles and abstracts, and then reviewed the full texts of eligible publications. In the event of a disagreement, they would discuss or solicit the opinion of the third reviewer until reaching a consensus.

### 2.2. Identification of Eligible Studies

Publications that met the following inclusion criteria were included in the systematic review and meta-analysis: (1) an experimental group given nusinersen or risdiplam treatment as an intervention for SMA; (2) a control group comprising SMA patients only given a placebo; (3) randomized controlled trials; (4) the main outcome indicators included at least one of the following: adverse events, the 32-item Motor Function Measure (MFM32), the Revised Upper Limb Module (RULM), Hammersmith Functional Motor Scale—Expanded (HFMSE), SMA Independence Scale–Upper Limb Module (SMAIS-ULM), motor-milestone response, Hammersmith Infant Neurological Evaluation Section 2 (HINE-2); and (3) sufficient data to calculate the odds ratio (OR) or weighted average difference (WMD).

The exclusion criteria were as follows: (1) studies with insufficient data; (2) letters, animal studies, protocols, conference abstracts, unpublished articles, or duplicated reports; (3) and single-arm studies.

### 2.3. Data Extraction

Two reviewers collected data independently, including, along with the summary and baseline from each study, the first author, publication year, country, registration of RCTs, study period, sample size, gender, age, SMA type, and follow-up duration. In addition, we also extracted outcomes related to milestones and adverse effects.

### 2.4. Quality Assessment

According to the Cochrane Risk of Bias tool (Cochrane Handbook for Systematic Reviews of Interventions, version 5.1.0) [23], we evaluated all the bias risks included in the RCT from the following aspects: random sequence generation, blindness of participants and personnel, blindness of result evaluation, incomplete result data, selective report, and other sources of bias. There are three grades in each aspect, including low risk, high risk, and unclear risk of bias. Any disagreement was resolved via a joint discussion with the third author.

### 2.5. Statistical Analysis

Review Manager (RevMan) software (version 5.4) (Cochrane Collaboration, Oxford, UK) was used for quantitative synthesis to evaluate treatment safety and effectiveness. We extracted the reported changes in the mean and standard deviation (SD) values of each motor function scale. Because of the various clinical features and rarity of SMA, the present data, which were derived from a limited number of RCTs, did not allow us to conduct a direct comparison of these drugs. The WMD and OR were calculated for the comparison of continuous and dichotomous variables with the same outcome measures, respectively, with 95% confidence intervals (CIs) determined in a meta-analysis model. Subsequently, the heterogeneity in studies was assessed through the chi-square test, and the I-square test was used to quantify its extent. A Chi-square *p* value < 0.05 or I^2^ > 50% indicated a lack of homogeneity of findings among studies. In addition, we also performed a one-way sensitivity analysis on the comprehensive results with significant heterogeneity to evaluate the impact of the inclusion studies. If there was significant heterogeneity, i.e., I^2^ > 50, a random-effects model was used; otherwise, a fixed-effects model was used [24]. The funnel chart was created with Review Manager version 5.3, and the publication bias was evaluated intuitively using Stata 15 (Stata Statistical Software: Release 15. College Station, TX, USA).

## 3. Results

### 3.1. Study Selection and Basic Characteristics

As illustrated in Figure 1, 310 relevant studies were retrieved, i.e., 29 studies from PubMed, 89 studies from Embase, 81 studies from Web of Science, and 111 studies from Cochrane Library. Then, 186 articles were included in the initial screening process for this study after eliminating duplications. After reading titles and abstracts, 166 articles were excluded based on the inclusion and exclusion criteria. Finally, of the 20 articles that remained, a total of 6 articles, comprising 6 RCTs enrolling 679 SMA patients, were included in the meta-analysis. The six RCTs [12,25,26,27,28,29] were double-blind and included 178 patients in the nusinersen group, 275 patients in the risdiplam group, and 226 patients in the placebo group. There were two phase II studies and four phase III studies. The information on the baseline characteristics of the included studies is shown in Table 1.

### 3.2. Quality Assessment of Inclusion

All trials were randomized, double-blind, and placebo-controlled. The results of the quality assessment and risk bias are summarized in Figure 2. Many studies were based on a small sample size of participants, and some variation was found amongst the trial results as well as the study design, which would increase the risk of bias, but this does not indicate that the studies were of low quality.

### 3.3. Effects of Interventions

#### 3.3.1. Change in HFMSE

A total of three studies including 472 patients (84 nusinersen and 237 risdiplam versus 151 placebo patients) reported HFMSE results [25,26,29]. One study [29] showed the HFMSE score was increased after nusinersen treatment (WMD: 4.90; 95% CI: 3.17, 6.63; *p* < 0.00001). In addition, the pooled analysis between risdiplam and the placebo showed a statistically significant improvement in HFMSE (WMD: 0.87; 95% CI: 0.05, 1.68; *p* = 0.04), and no significant heterogeneity was observed (I^2^ = 0%, *p* = 0.46). The difference between the two subgroups was significant (*p* < 0.00001) (Figure 3A).

#### 3.3.2. Change in MFM32

The two RCTs with sufficient data for an analysis of the MFM32 score, including 336 patients (227 risdiplam versus 109 placebo patients) [25,28], showed risdiplam significantly increased the MFM32 score at the treatment endpoint compared with the placebo (WMD:1.48; 95% CI: 0.58, 2.38; *p* = 0.001). The heterogeneity test showed no significant difference (I^2^ = 0%, *p* = 0.87) (Figure 3B).

#### 3.3.3. Change in RULM

Three studies with a total of 446 patients showed a change in RULM score [12,25,26], including one evaluated nusinersen treatment (66 nusinersen and 235 risdiplam versus 145 placebo patients) [12]. There was a significant increase in RULM score after the nusinersen treatment (WMD: 3.70; 95% CI: 3.30, 4.10; *p* < 0.00001). At the same time, the other two studies including risdiplam treatment also produced a significant improvement (WMD: 1.29; 95% CI: 0.57, 2.01; *p* = 0.0005), and there was no significant heterogeneity among the studies (I^2^ = 0%, *p* = 0.42). Subgroup analysis showed a significant difference between groups (*p* < 0.00001) (Figure 3C).

#### 3.3.4. Change in HINE-2

Analysis of HINE-2 score was conducted in two studies with only 58 patients (40 nusinersen versus 18 placebo patients) [12,27]. The pooled analysis showed a statistically significant improvement in HINE-2 (WMD: 5.21; 95% CI: 4.83, 5.60; *p* < 0.00001), and no significant heterogeneity was observed (I^2^ = 0%, *p* = 0.84) (Figure 3D).

#### 3.3.5. Change in SMAIS

Data on SMAIS were synthesized from two studies including 341 patients (228 risdiplam versus 113 placebo patients) [25,26]. The results of the pooled analysis showed no significant difference in the SMAIS score between the risdiplam and placebo groups (WMD: 1.31; 95% CI: −1.10, 3.72; *p* = 0.29) with statistically significant heterogeneity (I^2^ = 79%, *p* = 0.03) (Figure 3E).

#### 3.3.6. Safety

All six RCTs [12,25,26,27,28,29] reported data from the included studies on total adverse events (AEs) and severe adverse events (SAEs), including 679 patients (178 nusinersen and 275 risdiplam versus 226 placebo patients). According to the incidence of AEs, there was no significant difference between the risdiplam and placebo groups (OR: 1.08; 95% CI: 0.55, 2.13; *p* = 0.82) with non-significant heterogeneity (I^2^ = 0%, *p* = 0.79). Meanwhile, participants treated with nusinersen, compared with those treated with a placebo, did not show a significant difference in AEs (OR: 0.57; 95% CI: 0.15, 2.14; *p* = 0.22) without significant heterogeneity (I^2^ = 33%, *p* = 0.22). Subgroup analyses showed no significant difference (*p* = 0) (Figure 4A).

When considering the incidence of SAEs in the risdiplam and placebo groups, the pooled effect estimates showed no significant difference (OR: 1.11; 95% CI: 0.66, 1.88; *p* = 0.7) with non-significant heterogeneity (I^2^ = 0%, *p* = 0.96). Similarly, no significant difference in SAEs was observed between the nusinersen and placebo groups (OR: 0.54; 95% CI: 0.22, 1.30; *p* = 0.17) without significant heterogeneity (I^2^ = 47%, *p* = 0.15). Subgroup analyses showed no significant difference (*p* = 0.17) (Figure 4B).

#### 3.3.7. Sensitivity Analysis

One-way sensitivity analyses were conducted to compare the scores for HFMSE, RULM, and SMAIS and severe adverse events, so as to evaluate the impact of each single study on the results to demonstrate stability and sensitivity by removing each study one by one. The results suggest that the analysis was reliable and stable (Appendix A). When we excluded the data of the open-label extension study by Finkel et al. conducted in 2017, the heterogeneity of SAEs disappeared (I^2^ = 0%, *p* = 0.86), indicating that most of the heterogeneity was explained by this study.

#### 3.3.8. Publication Bias

We carried out Egger’s linear regression test and funnel plot method to investigate the potential publication bias in our meta-analysis. Visual inspection of funnel plots for the outcomes did not show distinct asymmetry, indicating a good stability of results. (Appendix A).

## 4. Discussion

To our knowledge, there are only three drugs approved for the treatment of SMA, including nusinersen, onasemnogene abeparvovec, and risdiplam. However, there have been no RCT studies on onasemnogene abeparvovec treatment so far. Our meta-analysis included data from all reported RCTs. It is the first of its kind to our knowledge, and aimed to investigate the efficacy and safety of nusinersen and risdiplam in the treatment of 728 SMA patients. A previous meta-analysis only compared the safety and efficacy of nusinersen with a placebo [30], and some studies reported an improvement in motor and pulmonary function tests in SMA patients post nusinersen treatment based on real-world observational data [31,32]. Unlike these meta-analyses, we added three high-quality RCTs to evaluate the risdiplam treatment of SMA and performed sensitivity analyses to make the results more credible. Our study observed that the nusinersen group achieved clinical meaningful motor function based on the scores of the HFMSE, RULM, and HINE-2 compared with the control group. At the same time, the reliable data of included studies on risdiplam also showed a significant improvement in the HFMSE, MFM32, RULM, and SMAIS scores, indicating its benefit for further improvement or stabilization of motor function.

In the majority of the studies, the HFMSE and RULM were the functional measures most commonly used to assess efficacy, followed by the MFM32 and, less frequently, by the HINE2 and SMAIS. The HFMSE consists of 33 items representing upper and lower extremity motor function, with scores reaching up to 66 points [33], while the RULM focuses on upper limb function, reaching up to 37 points [34]. The MFM32 is a 32-item scale and measures three domains: standing transfers and ambulation, proximal and axial, and distal function [35]. In addition, the SMAIS was developed to assess the level of assistance required for individuals with type 2 and non-ambulant type 3 SMA to perform typical daily activities [36], while HINE-2 is a test that evaluates the ability of infants or young children to perform different activities that require using muscles [37]. On all the motor function scales used above, higher scores indicate better motor function. Despite the promising efficacy results in our study, none of the SMA patients attained normal motor development, and a large proportion of the patients still presented with severe clinical manifestations.

Apart from the efficacy of interventions, it is also important to consider their safety and tolerability. As far as we know, nusinersen cannot penetrate the blood–brain barrier (BBB), and it can only be administrated into the spinal cord via intrathecal injections to display its treatment effect [38]. In fact, it is a valuable treatment option that is currently used for the treatment of adult patients with 5q SMA [31]. Complications of performing lumbar puncture in SMA patients include headache, backache, infection, CSF leakage, and exacerbating respiratory function impairment. In addition, repeated lumbar punctures can present challenges in some chronic SMA patients, especially those with significant scoliosis when modern imaging assistance is lacking [39]. These three trials reported that the most common AEs noted in the nusinersen group were pyrexia, constipation, and upper respiratory tract infections [12,27,29]. Additionally, risdiplam can penetrate the BBB, increasing the level of SMN protein in both the central nervous system and peripheral organs through oral administration. The most common AEs in patients receiving risperidone treatment include fever, diarrhea, infections, and pneumonia. Since some studies did not report these events in detail, we did not compare the incidence of AE alone. In our analysis, there were no significant differences between the treatment group and placebo group in terms of any adverse effects and serious adverse effects.

The present study reports the latest and largest evidence-based analysis conducted so far, and evaluates SMA patients who received nusinersen and risdiplam treatment. All RCTs in our research are considered to be of high quality. However, at present, we have to admit several limitations of the study. Firstly, the popularization of our results could be biased by the small number of available trials and the relatively small number of patients, which reduced the overall quality of evidence for this meta-analysis and did not allow us to conduct a classic systematic review with an indirect comparison with network. Secondly, the average age of the patients included in this study fluctuated between 163 days and 18.5 years old, and not all the studies included in this study evaluated patients with the same characteristics, which resulted in differences in SMA types, clinical symptoms, and disease severity among patients in the final analysis results. Thirdly, we compared these two drugs separately with a placebo, without comparing the efficacy of these two drugs, due to the lack of direct comparative RCTs. For SMA children who do not meet the selection criteria and adults experiencing SMA, the benefits and risks of nusinersen are uncertain. Fourthly, we only included randomized clinical trials in this systematic review, using strict inclusion criteria, which may have led to the occurrence of selection bias in studies related to rare diseases, such as 5q SMA. Finally, we used the HFMSE, MFM32, RULM and HINE-2 as SMA motor function assessment scales, which may not be sensitive enough to evaluate the therapeutic efficacy in SMA patients with different SMA types or different clinical manifestations that have been previously reported [40]. Due to the lack of reported data for cerebrospinal fluid or serum biomarkers, we did not include these to assess efficacy, although they have recently become increasingly important in most recent trials involving antisense oligonucleotides and gene therapies [41]. Therefore, there is publication bias. In general, these results show that nusinersen and risdiplam have a positive influence on the disease progression of patients with SMA, but their curative effect on the symptoms of functional, behavioral, and other systemic change in patients with later-onset SMA is doubtful.

A previous study on SMA type I patients who were treated with nusinersen showed that HINE-2 scores were inversely correlated with disease duration [42]. In the ENDEAR trial, infants with SMA who received nusinersen treatment were more likely to show improvement and had a motor-milestone response (based on the HINE-2 score) that increased over time [29]. In another study, SMA type 2 and type 3 patients, who had a longer disease duration, demonstrated a poor treatment response [43]. As a result, early diagnosis and shorter disease duration may be predictors of better treatment outcomes with nusinersen. Our meta-analysis indicates that nusinersen and risdiplam are generally well-tolerated and effective treatments in SMA patients. Future investigations should be focused on stratifying patient categories, including ages and rating scales, and should take into account specific situations with better-defined methods, inclusion criteria, and outcome measures, including cost-effectiveness and self-evaluated scales in different aspects.

## 5. Conclusions

Based on the evidence currently available in our study regarding drug treatment, nusinersen and risdiplam were effective in the treatment of patients with SMA, and both groups were similar in terms of adverse effects and serious adverse effects when compared with the control group. However, there are no specific studies that have conducted a comparison between the drugs nusinersen and risdiplam. Future well-designed RCTs are needed to consider broader identical inclusion and exclusion criteria, longer follow-up periods, and assessment measures with more accuracy. Moreover, new population studies are needed in the future to better determine the optimal use of each drug in SMA patients. We should also focus on developing other novel disease-modifying therapies for patients with SMA of all ages and severities.

## Figures and Tables

**Figure 1 brainsci-13-01419-f001:**
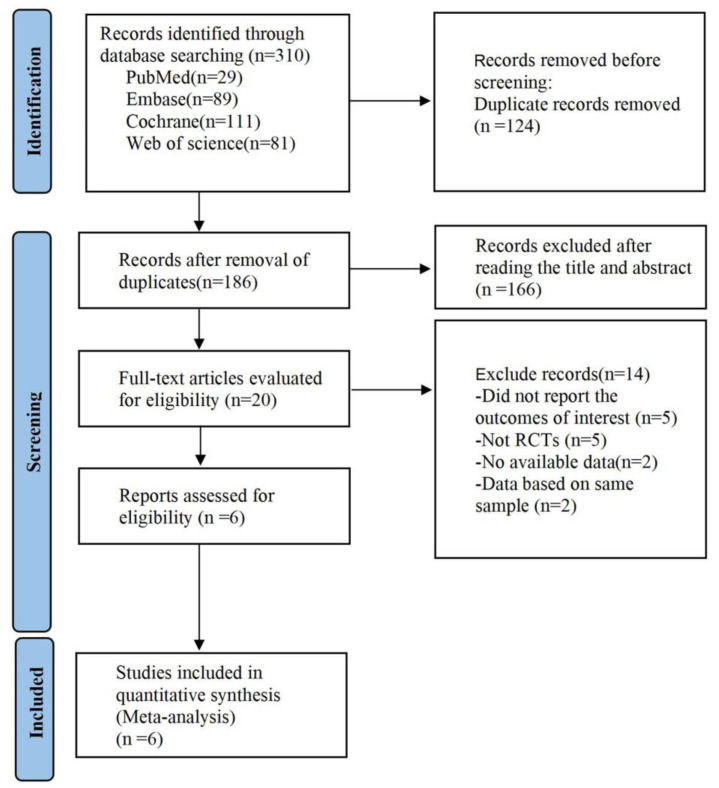
Flowchart of study selection process for meta-analysis.

**Figure 2 brainsci-13-01419-f002:**
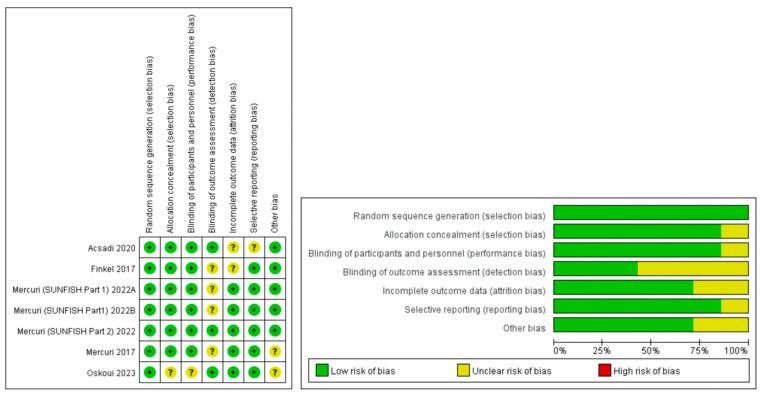
Quality assessment of the risks of biases in the studies in the meta-analysis.

**Figure 3 brainsci-13-01419-f003:**
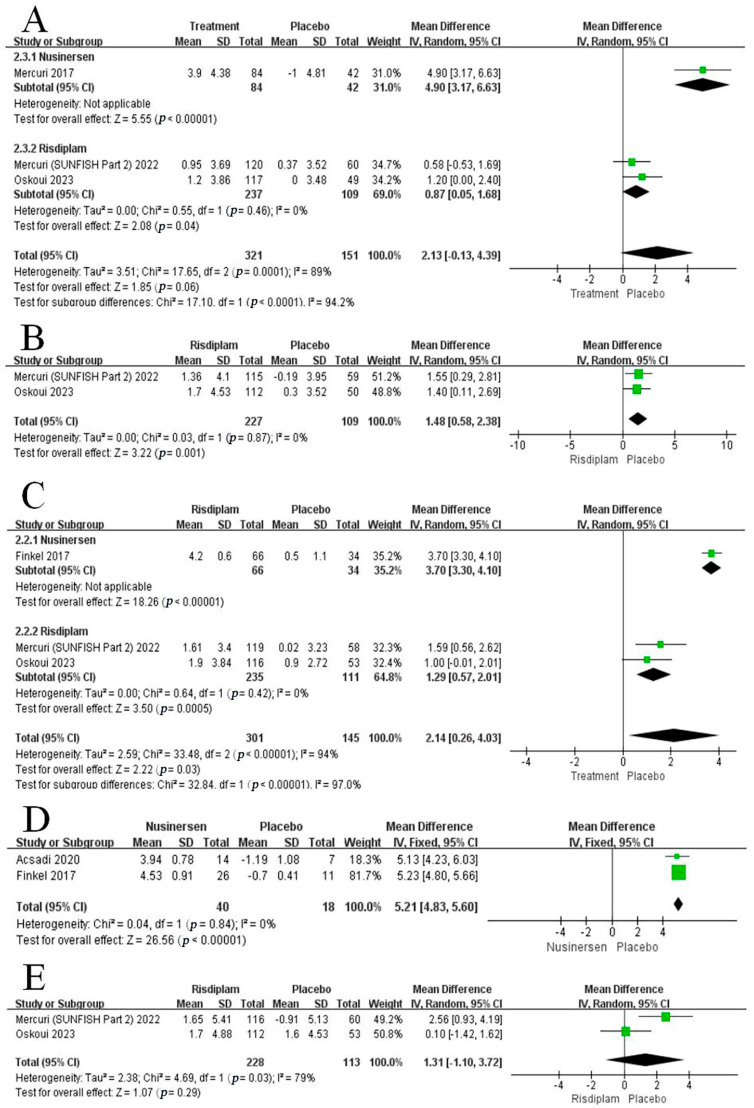
Forest plot of mean difference (MD) in (**A**) HFMSE, (**B**) MFM32, (**C**) RULM, (**D**) HINE-2, and (**E**) SMAIS.

**Figure 4 brainsci-13-01419-f004:**
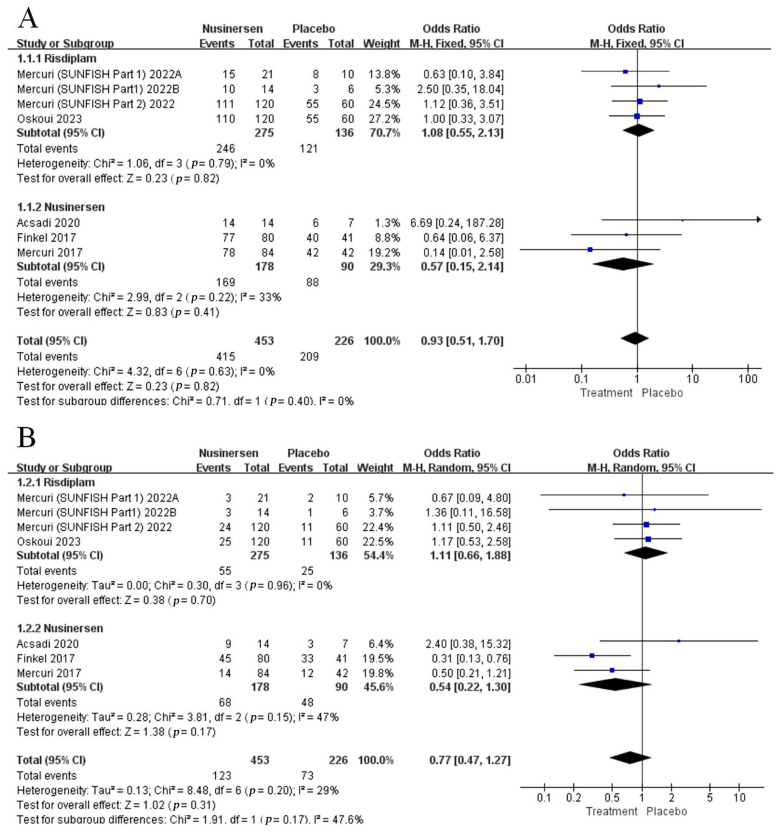
Forest plot of safety endpoints: (**A**) adverse events and (**B**) severe adverse events.

**Table 1 brainsci-13-01419-t001:** The characteristics of the studies selected for the meta-analysis.

Study	Country	Register Number	Phase	Intervention	Gender (F/M)	Age, Mean Years (SD)	SMA Type	MFM32 Total Score, Mean (SD)	RULM Total Score, Mean (SD)	HFMSE Total Score, Mean (SD)	HINE-2score, Mean (SD)	Follow Up	Outcome
Mercuri 2022A	Italy	NCT02908685	II	Group A: Risdiplam: 21	14/17	5 (6.7)	3–4	44.4 (11.9)	-	-	-	12 weeks	F1, F2
				Group A: Placebo: 10					-	-	-		
				Group B: Risdiplam: 14	13/7	14.5 (8.9)	2–4	40.9 (18.2)	-	-	-		
				Group B: Placebo: 6					-	-	-		
Mercuri 2022B	Italy	NCT02908685	III	Risdiplam: 120	61/59	9 (17.04)	2–3	45.48 (12.09)	19.65 (7.22)	16.10 (12.46)	-	12 months	F1, F3, F4, F5, F7
				Placebo: 60	30/30	9 (16.30)		47.35 (10.12)	20.91 (6.41)	16.62 (12.09)	-		
Oskoui 2023	Canada	NCT02908685	III	Risdiplam: 120	60/55	10 (17.03)	2–3	45.5 (12.7)	-	-	-	12 and 24 months	F1, F2, F3, F4, F5, F7
				Placebo: 60	62/52	8 (19.26)		46.2 (13.0)	-	-	-		
Acsadi 2020	United States	NCT02462759	II	Nusinersen: 14	5/9	16.7 (30.59)	2–3	-	-	-	7.6 (5.4)	14 months	F1, F2, F6
				Placebo: 7	5/2	18.5 (28.15)		-	-	-	5.9 (4.5)		
Mercuri 2017	Italy	NCT02292537	III	Nusinersen: 84	46/38	3 (3.7)	2–3	-	19.5 (6.2)	22.4 (8.3)	-	15 months	F1, F2, F4, F5
				Placebo: 42	21/21	4 (5.2)		-	18.4 (5.7)	19.9 (7.2)	-		
Finkel 2017	United States	NCT02193074	III	Nusinersen: 80	43/37	163 (47.5) days	1	-	-	-	1.29 (1.07)	394 days	F1, F2, F6
				Placebo: 41	24/17	181 (58) days		-	-	-	1.54 (1.29)		

F1: AE, any adverse event; F2: SAE, severe adverse event; F3: change in MFM32 total score; F4: change in RULM total score; F5: change in HFMSE total score; F6: change in HINE-2 score; F7: change in SMAIS score; SD, standard deviation.

## Data Availability

The generated or analyzed data presented in the study are included in the article/Appendix A; further inquiries can be directed to the corresponding author.

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
