# Peer review of "Safety and Efficacy of Nusinersen and Risdiplam for Spinal Muscular Atrophy: A Systematic Review and Meta-Analysis of Randomized Controlled Trials"

_brainsci, 2023, doi:10.3390/brainsci13101419_

Round 1

Reviewer 1 Report

The systematic review and meta-analysis of the efficacy and safety of nusinersen and risdiplam in the treatment of SMA addresses a topical issue of major importance for medical specialists and patients with SMA. The databases analyzed are recognized for the quality of their published work and therefore represent significant sources for the information processed in your manuscript.

The Introduction section presents sufficient information for the topic addressed in the paper, i.e., definitions, etiology, classification, and currently existing therapeutic means.

The Methods section describes in detail how the literature review was conducted, how eligible studies were selected, how data were extracted, and how they were processed.

The results are clearly, accurately, and thoroughly presented.

However, in subchapter 3.3.6 Safety (lines 198-200), there appears to be a discrepancy in the number of patients mentioned. In this section, you refer to 679 patients in the six selected studies (275 nusinersen and 136 risdiplam versus 268 placebo), while in subchapter 3.1 Study selection and basic characteristics (lines 143-146), you mention 728 SMA patients (178 patients in the nusinersen group, 270 patients in the risdiplam group, and 280 in the placebo group). This discrepancy needs clarification to ensure the accuracy of the data.

The conclusions are clear and derive from the results presented.

Another information that needs to be clarified is the treatment with risperidone (line 18 in the abstract). Possibly this is a typo and should be corrected to "risdiplam" to accurately reflect the subject of the study.

Once these issues are addressed and clarified, I believe that your article is interesting and provides valuable information for both medical specialists and patients with SMA, meeting the criteria for publication. It is important to ensure that all data is consistent and accurate to maintain the scientific integrity of your research.

Author Response

Dear reviewer:

Thank you for your valuable remarks and suggestions. These remarks are very valuable and helpful for us to revise and improve the manuscript. We have studied the comments carefully and have made some amendments, hoping to get approval. The main corrections and replies to your opinions in the paper are as follows.

1. However, in subchapter 3.3.6 Safety (lines 198-200), there appears to be a discrepancy in the number of patients mentioned. In this section, you refer to 679 patients in the six selected studies (275 nusinersen and 136 risdiplam versus 268 placebo), while in subchapter 3.1 Study selection and basic characteristics (lines 143-146), you mention 728 SMA patients (178 patients in the nusinersen group, 270 patients in the risdiplam group, and 280 in the placebo group). This discrepancy needs clarification to ensure the accuracy of the data.

Response:

Thank you for your careful examination. According to the information, this section has been revised and modified. We checked and rewrote the sentences "The six RCTs were double-blind, including 178 patients in the nusinersen group, 275 patients in the risdiplam group and 226 patients in the placebo group.." (line149-152 ) in subchapter 3.1 Study selection and basic characteristics and “including 679 patients respectively (178 nusinersen and 275 risdiplam versus 226 placebo) in subchapter 3.3.6 Safety (line205-206 ). In addition, we modified the Figure 4 (A).

2. Another information that needs to be clarified is the treatment with risperidone (line 18 in the abstract). Possibly this is a typo and should be corrected to "risdiplam" to accurately reflect the subject of the study.

Response:

We deeply appreciate the reviewer’s suggestion. We rewrote the sentences “In addition, the risdiplam treatment group also showed statistically...” in abstract (line 19).

Reviewer 2 Report

1.     Please revise the term “risperidone” written in the abstract.

2.     All the abbreviations should be fully described in the first presentation. This should also be applied to the abstract.

3.     How did the authors compare different scales in their systematic reviews? How was this data standardized?

Author Response

Dear reviewer:

Thank you for your valuable remarks and suggestions. These remarks are very valuable and helpful for us to revise and improve the manuscript. We have studied the comments carefully and have made some amendments, hoping to get approval. The main corrections and replies to your opinions in the paper are as follows.

 1.Please revise the term “risperidone” written in the abstract.

Response:

We deeply appreciate the reviewer’s suggestion.

We rewrote the sentences “In addition, the risdiplam treatment group also showed statistically...” in abstract (line 19).

2. All the abbreviations should be fully described in the first presentation. This should also be applied to the abstract.

Response:

We thank the reviewer for pointing this out.

We added the abbreviations of the 32-item Motor Function Measure (MFM32), the Revised Upper Limb Module (RULM), Hammersmith Functional Motor Scale—Expanded (HFMSE), and Hammersmith Infant Neurological Evaluation Section 2 (HINE-2) in the first presentation in the abstract.

3. How did the authors compare different scales in their systematic reviews? How was this data standardized?

Response:

Thank you for your valuable remarks and suggestions.

Because the measurement methods of the motor scales used in each RCT were similar and did not cause large differences in the outcome variables, and the scales of each outcome indicator were the same, we present the calculated weighted mean differences (MDs) with 95% CIs for continuous outcomes. In addition, we extracted the change of scores from the baseline of each scales and compare the efficacy of nusinersen or risdiplam with placebo respectively. We added “We calculated the reported changes in mean and standard deviation (SD) values of each motor function scales” (line 129-131).

Reviewer 3 Report

The aim of the study was to conduct a systematic review and meta-analysis to evaluate the safety and effectiveness of nusinersen and risdiplam in the treatment of patients with spinal muscular atrophy. The authors conducted a systematic review and meta-analysis using PubMed, Embase, Web of Science and Cochrane.

1. In the Introduction section, the authors convincingly justify the need for research and the relevance of the work. This section did not raise any comments.

2. In the Materials and Methods section, the authors used a thoughtful algorithm for selecting and excluding publications, with the help of which several inclusion and exclusion criteria were determined. Next, the authors provide methodological algorithms for data extraction and quality assessment, as well as statistical analysis that was used in the work.

3. In the Results section, the authors presented their results in the form of compelling flowcharts in Figs. 1, as well as systematically processed data in Fig. 2, 3, 4. These sections of the results are written convincingly and do not cause any comments.

4. The Discussion section provides a competent and constructive discussion of the data obtained as a result of the meta-analysis. All stages of the discussion of randomized controlled trials are logically justified and allow for a substantive discussion of the obtained material, including a discussion of similar effects of drugs. I also consider this section to be satisfactory and not objectionable.

5. The conclusion of the work is convincing and consistent with the randomized studies conducted.

English language requires minor stylistic improvements

Author Response

Dear reviewer:

On behalf of all the contributing authors, I would like to express our sincere appreciations of your letter and reviewers’ constructive comments concerning our article entitled “Safety and Efficacy of Nusinersen and Risdiplam for Spinal Muscular Atrophy: A Systematic Review and Meta-Analysis of Randomized Controlled Trials” (Manuscript No: brainsci-2630572). Thank you for your valuable remarks and suggestions. These remarks are very valuable and helpful for us. 

Response:

Thank you for your valuable remarks and suggestions.

We checked our manuscript and corrected some grammatical and word errors.

Reviewer 4 Report

I would like to thank the Editor's invitation to review the manuscript entitled "Safety and Efficacy of Nusinersen and Risdiplam for Spinal Muscular Atrophy: A systematic review and meta-analysis of Randomized Controlled Trials". The authors presented a nice systematic review manuscript presenting the currently available evidence regarding the use of Nusinersen and Risdiplam in the treatment of patients with 5q SMA. The manuscript brings important data for the general reader, even for that one which is not a specialist in SMA and in Neuromuscular Diseases. Some points need to be carefully evaluated by the authors at this stage: 

1. There are some minor typos and technical aspects in the text to be evaluated by the authors at this time: 

- In the Abstract, in line 18, the authors described "risperidone" instead of "risdiplam". 

- All SMN1 and SMN2 genes descriptions in the manuscript must be presented with italics. I suggest authors to review mainly their Introduction and correct this aspect. 

- Line 230 - there is an unnecessary comma used in the sentence preceding "however" (I suggest to end the phrase and initiate a new sentence). 

- Line 261 - I suggest changing the expression "its curative effect" for a more appropriated meaning for the context.

2. As stated by the authors in some parts of the manuscript, I think it is important to describe in some sentences (as the content between lines 289-293) that there was an important limitation about the results of the study due to the stricter inclusion criteria and use only of randomized clinical trials in this systematic review (this may probably represent an important point to the occurrence of selection bias of studies related to rare diseases, such as 5q SMA). 

3. I think it would be of interest to mention in the text in the Discussion that systematic reviews which included not only randomized clinical trials identified important findings about the use of Nusinersen in adult patients with 5q SMA (Neurotherapeutics 2022;19(2):464-75). This is important to provide the correct with for the reader that this drug is in fact currently used for the treatment of adult patients with 5q SMA. 

4. The authors evaluated in their study mainly functional scales in adult patients with 5q SMA: they have not evaluated (or reported) data about cerebrospinal fluid or serum biomarkers taking into account the studies which were performed to date. This is an important aspect to be emphasized as biomarkers are expanding their importance in the most recent trials involving antisense oligonucleotides and gene therapies (Acta Neurol Belg 2023;123(5):1735-45). 

5. I suggest authors to highlight in their conclusion that the study showed that both drugs have good clinical results regarding efficacy and safety. However, there are currently no possibilities to compare the benefits or the impact on outcomes between the drugs Nusinersen and Risdiplam, as there are no specific studies which were performed to evaluate a comparison between groups. Certainly, in the future, there will be new population studies which will enable a better identification of the best profile of 5q SMA patients to use each one of the drugs. 

6. In the Discussion, I would put a special emphasis that not all the studies included in this systematic review have evaluated patients with the same profiles, as perfectly represented and summarized in Table 1. 

Round 2

Reviewer 2 Report

Dear Authors, I appreciate your comments. To clarify, I will try to provide a more detailed comment.

In Table 1, we have the scales that were used in the studies. The table shows that not all the studies used the same scales, which is concerning for systematic reviews.

Figure 3 shows the comparisons performed.

I will choose the first (A) comparison: Mercuri 2017, Mercuri (SUNFISH) 2022, Osokoui 2023.

i) Mercuri 2017 assessed RULM, HFMSE with Nusinersen

ii) Mercuri (SUNFISH) 2022 assessed MFM, RULM, HFMSE with Risdiplam

iii) Osokoui 2023 assessed MFM32 with Risdiplam

In this example, we do not have any available variables to compare following a standardized approach.

This can lead to misinterpretations of the clinical trials of these medications.

The authors can perform an indirect comparison with network. Noteworthy, this should be clearly stated in the study's abstract, methods, and limitations. However, the present data in the literature does not allow for a classic systematic review with metanalysis.

Author Response

Thank you very much for your comments and professional advice, these opinions help us to improve academic rigor of our article. Based on your suggestions and request, we have made corrected modifications on the revised manuscript, hoping to get approval. The main corrections and replies to the reviewer' opinions in the paper are as follows.

Question: In Table 1, we have the scales that were used in the studies. The table shows that not all the studies used the same scales, which is concerning for systematic reviews.

Figure 3 shows the comparisons performed.

I will choose the first (A) comparison: Mercuri 2017, Mercuri (SUNFISH) 2022, Osokoui 2023.

  1. i) Mercuri 2017 assessed RULM, HFMSE with Nusinersen
  2. ii) Mercuri (SUNFISH) 2022 assessed MFM, RULM, HFMSE with Risdiplam

iii) Osokoui 2023 assessed MFM32 with Risdiplam

In this example, we do not have any available variables to compare following a standardized approach.

This can lead to misinterpretations of the clinical trials of these medications.

The authors can perform an indirect comparison with network. Noteworthy, this should be clearly stated in the study's abstract, methods, and limitations. However, the present data in the literature does not allow for a classic systematic review with metanalysis.

Response:

We deeply appreciate your suggestion. Because of the limited data of nusinersen and risdiplam, we only performed a meta-analysis to compare the safety and efficacy of these drugs with placebo separately and we did not conduct the network meta-analysis. This is a limitation of our study. We rewrote the sentences “Further researches should provide a direct comparison between the two drugs in term of safety and efficacy.”(line 26-27) in the abstract, “Because of the various clinical features and rarity of SMA, the present data of limited randomized controlled trials (RCTs) does not allow to conduct directly comparison of these drugs.”(line 131-133) in method, and added “ and did not allow to conduct a classic systematic review with an indirect comparison with network” (line291-292) in discussion.